# Evaluation and Impact Mechanism of High-Quality Development in China’s Coastal Provinces

**DOI:** 10.3390/ijerph20021336

**Published:** 2023-01-11

**Authors:** Xiaojie Wang, Rongqing Han, Minghua Zhao

**Affiliations:** College of Geography and Environment, Shandong Normal University, No. 1 Daxue Road, University Science Park, Changqing District, Jinan 250358, China

**Keywords:** high-quality development, spatio-temporal evolution, driving mechanism, eastern coastal provinces

## Abstract

With economic expansion having moderated to a “new normal” pace, the eastern coastal provinces have been given a new historical task of high-quality development and become a window and frontier of China’s high-quality development. By designing and optimizing an index system of high-quality development levels and using the entropy-TOPSIS method, the study selected 21 indicators, include economic vitality, residents’ living standards, innovation efficiency and green development, and took China’s eastern coastal provinces as an example to study the characteristics of spatial-temporal variations in the high-quality development level from 2010 to 2020. Then, the study used the obstacle degree model to explore the factors that are obstacles to high-quality development. The results show that the high-quality development of the eastern coastal provinces presents an “up-down-up” fluctuation, with an increase of 40.1%. In particular, the development level of the residents’ living standards dimension is higher, and the high-quality development level of each province shows different degrees of growth and gradually tends to balanced development, with the high-quality development of Shanghai, Jiangsu Province and Zhejiang Province in a dominant position. The spatial pattern of high-quality development in the study areas shows a spatial distribution pattern of “high in the east and low in the west, high in the north and low in the south”, in which the bipolar spatial effect of the innovation efficiency dimension is becoming more and more prominent, while the regional synergistic development effect of the residents’ living standard dimension is more obvious, and the high-quality development spatial pattern shows a “core-periphery” structure, and there is a path-dependent effect in time change, and agglomeration is produced by trickle-down effect in space. The obstacles to residents’ living standards are high, and the main obstacle factor has gradually changed from insufficient output in innovation to a reduction in the scale of foreign trade. In addition, the problems of unreasonable industrial structure and shortage of per capita public cultural resources still exist. In provinces with a high-quality development level and a relatively developed economy, the biggest obstacle factors are economic vitality and residents’ living standards. In provinces with a low level of high-quality development and a relatively backward economy, the biggest obstacle factors are green development and innovation efficiency, and there are both similarities and differences in the main obstacle factors among provinces.

## 1. Introduction

At present, China’s economy is changing from a high-speed growth stage to a high-quality development stage. High-quality development is different from high-speed growth; it is efficiency- and quality-oriented, that is, reflecting quality first, efficiency first. The goal of high-quality development is to achieve higher quality, more efficient, more equitable and more sustainable development. This means that high-quality development cannot simply rely on the expansion of the quantity and scale of economic development, and refers to a sustainable development model that enhances comprehensive socio-economic competitiveness and focuses on ecological and environmental sustainability to meet the growing needs of the people for a better life [1,2]. However, as many regions are also faced with problems that cannot be ignored and need to be solved urgently, this means that economic development must change accordingly around the requirements of high-quality development and promote a new shift in economic construction of high quality. Measuring regional high-quality development levels, clarifying their temporal and spatial dynamic evolution characteristics and driving mechanisms, and continuing to give full play to regional advantages and leading roles are of great significance to the coordinated development of economy, society and ecological environment. 

The term “high-quality development” first appeared in the report of the 19th National Congress of China, but discussion on the connotation of high-quality development has not been unified. “High-quality development” essentially belongs to the scope of economic development. The early research was similar to that on “economic development, economic growth”. Now, the connotation analysis of high-quality development can be divided into single and multiple perspectives. Under the single perspective, scholars from different fields such as economy [3], society [4] and politics [5] have deeply discussed the connotation of high-quality development. Under multiple perspectives, most scholars agree that high-quality development is an efficient, fair and green sustainable development approach aimed at meeting people’s growing needs for a better life, and comprises the coordinated development of the “five-in-one” economic, political, cultural, social and ecological dimensions [6,7]. In fact, high-quality development involves more aspects and should be explored under a broader perspective. 

The evaluation of high-quality development is especially critical in the selection of indicators, and scholars have formed two views on the selection of indicators for the evaluation of high-quality development based on different understandings of the connotation of high-quality development: one is to measure high-quality development by total factor productivity [8,9,10]. In recent years, measuring high-quality development by total factor productivity has increasingly focused on environmental issues, and a number of scholars measured green total factor productivity as the basis for measuring high-quality development, thus reflecting the environmental issues involved in the efficiency of economic growth [11,12,13,14]. They not only consider the efficiency of factor inputs such as capital and energy in the production process and GRP output, but also add unexpected outputs such as wastewater output and SO_2_ emissions, thus reflecting the level of high-quality development more comprehensively. However, it is difficult to consider the comprehensive connotation of high-quality development by equating it with high-quality economic development. Meanwhile, another type of research has to do with constructing indicator systems based on the connotation of high-quality development, including the three-dimensional perspectives of economy, society, and ecological environment [15,16], the perspective of the five development concepts [17,18,19], the perspective of human–land coordination [20,21,22], and the perspective of economic development [23,24,25]. Compared with the two approaches, the latter focuses more on the multidimensional characteristics of high-quality development and is more in line with the connotation characteristics of high-quality development. 

Research on the factors influencing high-quality development is extensive, and scholars have explored the impact on regional high-quality development based on different perspectives under a single factor, under the interaction of multiple factors or under the mediating role, in order to provide countermeasure suggestions for promoting regional high-quality development from a specific area. Many scholars have found that human capital [26,27], globalization [28,29], environmental regulation [30,31], foreign investment [32,33,34], information and communication technologies [35,36,37], innovation [38,39], and finance [40,41,42] are factors that have an impact on the quality development of countries, provinces, regions and cities. In addition, some scholars have explored the impact of the interaction of multiple factors on high-quality development [43,44]. For example, by studying the relationship between energy consumption, FDI and economic development in Zhejiang Province, Zeng S concluded that foreign direct investment and energy efficiency promote each other, and that foreign investment indirectly contributes to high-quality development through energy use efficiency [45]. There are obvious regional differences in the scope of influence of each factor, the degree of influence, and the obstacle factors of regional development. Hence, each region should carry out analysis in accordance with its own geographical characteristics.

With the gradual deepening of reforms and market opening in China, its eastern coastal provinces are increasingly becoming the leaders of national economic growth, the frontiers of innovation and development, the front-runners of opening up, and the demonstration base for green development. These areas are typical of high-quality development in the new era and will effectively drive the high-quality development of the central and western regions onto the right track and provide strong support for the whole country along the high-quality development path [46,47,48]. Additionally, with the proposed “T” development strategy combining the eastern region and the Yangtze River basin, the eastern region has a more important role in strengthening regional leadership and promoting national regional coordination. However, there is still a considerable gap between advanced industrialization and the developed economies. In addition, the ability for innovation has not yet established a networked spatial pattern of regional linkage, and the rate of expansion of total import and export trade has slowed [49,50,51,52]. In the context of high-quality development, how to better play the leading role of the eastern region has become a key topic of concern for scholars.

Based on the above literature, the research on high-quality development has been enriched and expanded, but the literature on high-quality development and diagnosis of obstacle factors under the unified analysis framework is relatively rare. Therefore, compared with the existing literature, the contribution of this paper is as follows: (1) Based on multi-dimensional measurement of high-quality development levels, a relatively comprehensive and unified indicator system was formed to reflect the connotation of high-quality development. (2) The eastern coastal provinces serve as both administrative bodies with decision-making authority and early demonstration sites for high-quality development. Local administrators can consider the current status and evolution of their community’s economic, social, and resource environment development using the spatial and temporal evolution of their high-quality levels of development and the factors preventing it as a starting point for developing more effective development plans and policies.

The contents are as follows. First, we sort out the existing research on high-quality development. Second, by designing an index system of high-quality development, this paper measures the results of each dimension and comprehensive level of high-quality development in 11 coastal provinces (or cities) in China, and explores its temporal and spatial evolution characteristics and obstacle factors. Finally, the paper diagnoses the obstacle factors of high-quality development in 11 coastal provinces (or cities) in China and proposes targeted policy enlightenment.

## 2. Materials and Methods

### 2.1. Research Areas and Data Source

This paper focused on the eastern coastal provinces as the research area. The region includes nine provinces and two province-level municipalities (Figure 1). The eastern coastal provinces have always been the focus of China’s economic development. In 2020, the total GDP of the region was as high as CNY 53.692 trillion, accounting for 53.0% of the national GDP. The per-capita GDP was CNY 84,528 (data source: China Statistical Yearbook (2021)), which is much higher than the national average. The added value of the tertiary industry was CNY 29.689 trillion (data source: China Statistical Yearbook (2021)), accounting for 54.0% of the national total. The region’s local fiscal revenue was CNY 568.5467 billion (data source: China Statistical Yearbook (2021)), accounting for 57.6% of the national total. Retail sales of consumer goods came to CNY 2027.8520 billion (data source: China Statistical Yearbook (2021)), accounting for 50.6% of the national total. Total imports and exports of goods were valued at CNY 244,524.33 billion (data source: China Statistical Yearbook (2021)), accounting for 79.4% of the national total. According to the data, both in terms of the overall economic volume and the fraction of the population, the eastern coastal provinces have a clear advantage. As the hub of China’s population and economy, the eastern coastal provinces hold the most significant position in the country’s regional economic structure, and the levels of their economic development and the challenges they face in that regard have a direct bearing on the country’s overall social and economic progress. At the same time, local managers must develop effective policy measures in response to the current evolution of regional high quality development and development issues. High quality development is an important strategic plan for regional development, and how to choose a high quality development model suitable for the region’s own characteristics is a necessity. Therefore, taking the eastern coastal region as the research scale can provide corresponding policy inspiration and ideas for the coastal region’s high-quality development.

The research time frame for this paper was from 2010 to 2020. The social and economic data were from China Statistical Yearbook (2011–2021), China Urban Statistical Yearbook (2011–2021), China Environmental Statistical Yearbook (2011–2021) and the statistical yearbooks of 11 provinces and municipalities. The missing data in some years were supplemented by interpolation.

### 2.2. Construction of Evaluation Index System

Based on previous scholarly research, and taking into account the reliability, availability, and demand for quantifiable data, an evaluation index system reflecting high-quality development was constructed from the four dimensions of economic vitality, residents’ living standards, innovation efficiency and green development (Table 1). Specifically, economic vitality reflects the total economic volume and development power, which means that while maintaining the steady growth of GDP, the internal components of the economy, such as exports, consumption and industrial structure, are constantly optimized. The living standards of residents reflect that the concept of high-quality development is people-oriented and that development achievements are shared with the people, which means that social security, infrastructure and medical and educational levels are constantly improved, and the gaps between urban and rural areas are shrinking. Innovation efficiency reflects the kinetic energy and efficiency of high-quality development through the levels of science, education and experimental research. For green development, the “P-S-R” model is used to reflect the ecological environment in the process of development from the state of solid, liquid and gas pollutants [53,54,55].

### 2.3. Research Methods

(1) The entropy-TOPSIS method was used to evaluate the high-quality development level of China’s coastal provinces.

In this paper, the model effectively combined the entropy method and TOPSIS method [56,57,58], which had strong objectivity. The specific calculation method is as follows:

First, the raw data were standardized. In order to eliminate differences in the original data in terms of magnitude and attributes, standardization was used to deal with the original data. The calculation formula is as follows:$$\mathrm{Positive}\;\mathrm{index}:\;X_{\theta ij}=\frac{X_{\theta ij}-minX_{\theta ij}}{maxX_{\theta j}-minX_{\theta j}}.\;$$
$$\mathrm{Negative}\;\mathrm{index}:\;X_{\theta ij}=\frac{maxX_{\theta ij}-X_{\theta ij}}{maxX_{\theta ij}-minX_{\theta ij}}.\;$$
where *i* is the evaluation object, *j* is each evaluation object’s evaluation index, and *θ* is the year; therefore, *X_θij_* represents the standardization of the *j* evaluation indexes of the *i* evaluation object in the *θ* year, where *X_θij_* ∈ [0, 1].

Second, the entropy method was used for data weighting. The entropy method is a method of calculating the contribution of the index to the overall result, according to the variation degree of the evaluation index in different evaluation objects. The higher the degree of order of an evaluation index, the smaller the weight of the evaluation index, and vice versa. In order to meet the requirements for the base number in the subsequent logarithm calculation, the standardized data were translated by the amplitude A, and A = 1 in this paper. The calculation formula is as follows:
$$\mathrm{Determination}\;\mathrm{of}\;\mathrm{index}\;\mathrm{weight}:\;l_{\theta ij}=\frac{{X^{\prime }}_{\theta ij}}{\sum_{\theta }\sum_{j}{X^{\prime }}_{\theta ij}}.\;$$
$$\mathrm{Entropy}\;\mathrm{of}\;\mathrm{index}\;j:\;e_{j}=-k\sum_{\theta }\sum_{j}l_{\theta ij}\ln l_{\theta ij},k=\frac{1}{\ln\theta \cdot j}>0.\;$$
$$\mathrm{Calculation}\;\mathrm{of}\;\mathrm{weight}\;\mathrm{for}\;\mathrm{each}\;\mathrm{index}:\;w_{j}=1-e_{j}/\sum_{j}\left(1-e_{j}\right).\;$$
$$\mathrm{Calculation}\;\mathrm{of}\;\mathrm{contribution}\;\mathrm{degree}\;\mathrm{of}\;\mathrm{data}\;\mathrm{weighting}:\;Z_{\theta ij}=w_{j}\cdot X_{\theta ij}\;.\;$$

Third, the closeness was measured by the TOPSIS method. TOPSIS is a method of sorting the advantages and disadvantages by calculating the distance between the target data and the optimal and worst schemes. The higher the closeness of an evaluation object, the higher the high-quality development level of the region, and vice versa. In this study, the order of closeness was used to express the differences in high-quality development level among provinces and cities. The closeness was between 0 and 1. The calculation formula is as follows:$$\mathrm{Optimal}\;\mathrm{scheme}:\;z_{\theta j}^{+}=\underset{\begin{array}{c} 1\leq \theta \leq 10\\ 1\leq j\leq 11 \end{array}}{\max}z_{\theta j}.\;$$
$$\mathrm{Worst}\;\mathrm{scheme}:\;z_{\theta j}^{-}=\underset{\begin{array}{c} 1\leq \theta \leq 10\\ 1\leq j\leq 11 \end{array}}{\min}z_{\theta j}.\;$$

Euclidean distance between the evaluation object and the optimal scheme: $$d_{i}^{+}=\sqrt{\sum_{i}{\left(z_{\theta ij}-z_{\theta j}^{+}\right)}^{2}}.\;$$

Euclidean distance between the evaluation object and the worst scheme: $$d_{i}^{-}=\sqrt{\sum_{i}{\left(z_{\theta ij}-z_{\theta j}^{-}\right)}^{2}}.\;$$
$$\mathrm{Calculate}\;\mathrm{the}\;\mathrm{nearness}\;\mathrm{degree}:\;v_{i}=d_{i}^{-}/\left(d_{i}^{+}+d_{i}^{-}\right).\;$$

(2) The coefficient of variation was used to measure the differences in high-quality development level among regions in the eastern coastal provinces.

In this paper, the coefficient of variation [59,60] was used to compare the degree of variation among regions in each year. The greater the coefficient of variation, the more different the high-quality development level among regions, and vice versa. The specific calculation method is expressed as follows:$$C=\frac{\sigma_{j}}{\mu_{j}}$$
where *i* is the evaluation object, *σ_j_* is the standard deviation in the high-quality development level of *j* research objects, and *μ_j_* is the average value of the high-quality development level of *j* research objects. 

(3) The obstacle model was used to calculate the barrier factors and barrier degrees.

In this paper, the obstacle model [6] was used to calculate the barrier factors and barrier degrees of high-quality development in the eastern coastal provinces altogether. The greater the degree of an obstacle, the stronger the obstacle to the high-quality development of the region. Hence, to explore the main factors restricting the high-quality development of the region, the specific calculation method is expressed as follows: $$M_{ij}=\frac{F_{j}\cdot O_{ij}}{\sum_{i=1}^{n}F_{j}\cdot O_{ij}}\times 100\%$$
where *F_j_* represents the contribution degree of the factor, that is to say, the influence degree of the factor on the overall goal; *O_ij_* represents the difference between the single factor and the maximum scheme, expressed by 1 *− X_ij_*,

## 3. Evaluation of High-Quality Development Level in China’s Coastal Provinces

### 3.1. Temporal Evolution of the Level of Quality Development in the Study Area

#### 3.1.1. Temporal Evolution of Overall High-Quality Development Level

Using the entropy-TOPSIS model, we calculated the evolution of the temporal pattern of high-quality development level in the eastern coastal provinces of China from 2010 to 2020. The overall high-quality development level of the eastern coastal provinces showed an upward “rise-decline-rise” fluctuation (Figure 2).

The level of quality development in coastal provinces showed steady growth from 3.212 in 2010 to 4.979 in 2020, an increase of 55.0%. The changes in the level of quality development throughout the study period can be divided into three stages. In the first stage, the comprehensive score for high-quality development level increased from 3.212 to 3.977 from 2010 to 2012, for an increase of 23.8%, and the average annual growth rate was 61.9%, which indicated a period of rapid growth. In the second stage, the total score for high-quality development level decreased slightly, by 0.113 from 2012 to 2014, for a decrease of 2.8%, and the average annual rate of decline was 61.9%, marking a period of slow decline. In the third stage, the level of high-quality development increased steadily, by 1.115 from 2014 to 2019, for an increase of 28.9%, and the average annual growth rate was 21.5%, indicating a period of steady growth. 

#### 3.1.2. Temporal Evolution of the Four Dimensions of High-Quality Development Level

The study calculated the evolution of the temporal pattern of the high-quality development level in terms of the four dimensions of economic vitality, residents’ living standards, innovation efficiency and green development in the eastern coastal provinces from 2010 to 2020 (Figure 3). The overall development level of each dimension has improved significantly, while the change in growth of each dimension has obvious differences, among which innovation efficiency has grown the most, followed by the living standards of residents, and the evaluation value is significantly higher than for other dimensions. The specific changes are as follows:

Economic vitality rose from 0.884 in 2010 to 1.375 in 2020, an increase of 55.5%, showing a continuous growth trend, and the overall economic vitality of the eastern coastal region developed steadily and positively.

Residents’ living standards rose from 1.504 in 2010 to 2.440 in 2020, an increase of 62.3%, and the evaluation value showed the characteristics of an inverted “V” from 2010 to 2014. Statistical data indicate that in 2014, the values of “public libraries per capita”, “urban water usage”, and “road area per capita” in Guangdong Province, Guangxi Province, and Fujian Province fell. The evaluation value showed a high growth trend, reflecting the continuous improvement in objective indicators such as health care coverage, infrastructure, and living environment in all provinces and municipalities, and the increasing quality of economic development contributed to the improvement in the actual quality of life of the people.

Innovation efficiency increased from 0.860 in 2010 to 1.428 in 2020, for a maximum rise of 66.1%, showing a year-on-year rise, but the annual growth rate kept fluctuating and decreasing, compared with the 22.6% growth rate from 2010 to 2012. The growth rate in evaluation value in 2018–2020 was only 10.1%, reflecting that sustained growth in scientific and technology innovation capacity encountered a bottleneck.

Green development increased from 1.032 in 2010 to 1.204 in 2020, with a minimum increase of 16.6%. Meanwhile, it decreased from 1.165 to 1.158 during 2016–2018. This shows that in the future, provinces and cities in the coastal region should accelerate the green and low-carbon transformation, improve the efficiency of resource and environmental utilization and environmental governance, reduce pollutant emissions, and continuously optimize the pattern of coordinated regional development.

#### 3.1.3. Temporal Evolution of High-Quality Development Level by Provinces and Cities

The study calculated the evolution of the temporal pattern of the high-quality development level of provinces and cities in the eastern coastal region (Figure 4 and Table 2). The provinces and cities in the eastern coastal region showed different degrees of growth in the level of high-quality development, but the ranking of the provinces and cities was relatively stable and gradually tended toward balanced development. 

From the regional differences in high-quality development, the coefficient of variation of the evaluation value of high-quality development in the eastern coastal provinces decreased by 0.06, for a decrease of 18.0%, and the range increased by 0.05, for an increase of 14.4%. This indicates that the measures of dispersion of high-quality development in the eastern coastal provinces were decreasing, but the development gap between the areas with the highest and lowest development levels was still increasing.

First of all, Shanghai City, Jiangsu Province and Zhejiang Province have always been in a dominant position. With its superior development foundation, Shanghai City has become the leader of high-quality development in the eastern coastal provinces. However, its growth rate was only 28.6%, and the improvement in its economic development quality was the slowest. Compared with Shanghai City, the high-quality development levels of Zhejiang Province and Jiangsu Province have increased significantly and have great development potential. 

Secondly, the positions of Guangdong Province, Tianjin City, Shandong Province and Fujian Province were in the middle. Among them, with its continuous improvement in innovation efficiency and residents’ living standards, Fujian Province had a high-quality development level that increased by 63.7%, ranking third. Compared to Fujian Province, Guangdong Province, Shandong Province and Tianjin City had modest growth in their high-quality development levels, at 58.0%, 44.7% and 52.0%, respectively. The reason is that there were obvious weaknesses in high-quality development in these provinces. Specifically, Tianjin City still had serious deficiencies in infrastructure, social security and balanced urban and rural development. Determining how to ensure that high-quality development achievements benefit the people and improve their happiness is a problem that cannot be ignored. In terms of economic foundation, industrial structure and foreign trade, Guangdong Province still had a large gap with the other strong developing provinces. Economic vitality has become the key factor limiting high-quality development in Guangdong Province. In addition, the level of the green development dimension showed a downward trend in Shandong Province. The key issues for high-quality development in Shandong Province are how to promote environmental protection and construction, comprehensively improve the level of pollution control, promote the continuous improvement of environmental quality, and realize the continuous promotion of green development to high-quality development.

Finally, high-quality development in Hainan Province, Hebei Province, Guangxi Province and Liaoning Province was at a low level. Among them, the high-quality development level in Liaoning Province increased from 0.18 to 0.34, for an increase rate of 88.4%, ranking it first, which indicated that economic revitalization and development had entered a new stage in Liaoning Province. Compared with Liaoning Province, the growth rates of high-quality development in Hainan Province, Hebei Province and Guangxi Province were small, at 56.7%, 54.2% and 49.3% respectively. The reason is that Hainan Province and Guangxi Province had small growth rates in the dimension of innovation efficiency, and Hebei Province had a small growth rate in the dimension of residents’ living standards; these have become factors restricting their respective levels of high-quality development.

### 3.2. Spatial Evolution of the Level of Quality Development in the Study Area

#### 3.2.1. Spatial Evolution of Overall High-Quality Development Level

Using the Geostatistical Analyst in ArcGIS, the study illustrated the spatial evolution of high-quality development in each direction in the eastern coastal provinces from 2010 to 2020 (Figure 5). In general, the spatial distribution pattern for coastal provinces was relatively stable, showing a spatial distribution pattern of “high in the east and low in the west, high in the north and low in the south”.

The level of high-quality development in the eastern coastal provinces has been relatively flat in both the north–south and east–west directions, with no major fluctuations or ups and downs. The trend line in the north–south direction has always maintained the characteristic of an inverted “U” distribution, and has obviously been higher in the south than the north, and the convex points have been mainly located in Jiangsu Province and Shanghai. At the same time, the gap between the extreme points of this curve increased significantly during the study period, mainly showing a significant increase in the extreme values in the north–south direction and a smaller increase in the range of the extreme small values, such that the gaps between high-quality development in the north–south direction were significantly enhanced. In addition, the trend line in the east–west direction always maintained a high east–west development trend. Hence, the eastern regions of Shanghai, Jiangsu Province, and Zhejiang Province were high-value areas of high-quality development, the western region of Guangxi Province was a low-value area of high-quality development, and the central region connected the east and west with a gentle transition.

#### 3.2.2. Spatial Evolution of the Four Dimensions of High-Quality Development Level

To determine the spatial pattern of evolution of the development level of each dimension in the eastern coastal provinces from 2010 to 2020, the evaluation values of these four dimensions were normalized from 0 to 1, and the processed evaluation values were divided into five stages from high to low at equal intervals (Figure 6). In general, the spatial evolution of each dimension in the eastern coastal provinces was relatively stable, and there was a significant increase in the development level of each dimension, showing a spatial distribution pattern with high value areas as the core in general.

The high-value area of economic vitality was Shanghai, which had a dominant position in terms of economic base, industrial structure, foreign trade, etc. and gave full play to the feeder role of the central city, driving a continuous increase in the economic vitality of its surrounding provinces, including Jiangsu Province and Zhejiang Province, forming a cluster of high-value economic vitality. In addition to Jiangsu Province and Zhejiang Province, Guangdong Province and Hainan Province were also in the higher value area of economic vitality development. However, due to the natural environment and historical reasons, the radiated drive to the surrounding areas was weak and useless in terms of forming a strong spatial spillover effect.

The high-value areas of residents’ living standards were Jiangsu Province and Shanghai City, where better economic foundations provided the basis for the development of infrastructure, education and medical care, and cultural life, transforming the fruits of development into people’s welfare and improving the quality of development and governance. In general, a more balanced development pattern was formed, mainly manifested in the priority development of the northern coastal provinces, gradually driving the southern provinces to achieve an overall improvement in the living standards of residents.

The high-value areas of innovation efficiency were Guangdong Province, Jiangsu Province and Zhejiang Province, among which Jiangsu Province and Zhejiang Province brought together various elements of enterprises, universities and R&D institutions to promote the construction of innovation clusters and become innovation hubs in the Yangtze River Delta region, together with Shanghai. In contrast, the synergistic linkage effect between Guangdong Province and its periphery was not obvious; there was a big gap with the surrounding areas, and the “barrel effect” was significant. Therefore, Guangdong Province, on the basis of self-improvement, needs to consider how to become a powerful engine of innovation development, foster the spread of innovation development into its surrounding areas, and provide a strong impetus for regional innovation development, with more obvious spatial differences and significant polarization trends in general.

The high-value areas of green development were more concentrated, mainly in the northern Tianjin, central Jiangsu Province, Shanghai, Zhejiang Province agglomeration, as well as in southern Guangdong Province. In general, the overall concentration of the continuous stratified agglomeration phenomenon was significant and constantly trended toward balanced development. However, in Liaoning Province and Guangxi Province, the industrial structure was more backward, with concentrations of highly polluting and energy-consuming industries, and green development was at a lower level.

#### 3.2.3. Spatial Evolution of High-Quality Development Level by Province

The evaluation value of high-quality development levels was subject to standardization that distributed it between 0 and 1. Subsequently, the treated evaluation value was divided into five stages, from high to low. There were spatial differences in the high-quality development levels of the studied provinces from 2010 to 2020 (Figure 7). The development levels showed the characteristics of “center–periphery”. The “center” was Shanghai City and Zhejiang Province, and the high-quality development levels of the other provinces were low. From the perspective of the development process, the number of provinces in different, high-quality development stages has changed significantly (Table 3). The numbers of median areas, higher areas and high areas increased to varying degrees, and the number of lower area and low areas decreased significantly. This shows that the eastern coastal provinces have made great progress in various fields, such as economy, society and ecological environment, and the level of high-quality development has been significantly improved.

From the perspective of spatial pattern, there was a circular cumulative effect (CCE) in areas of high economic development level with Shanghai City as the core. On the one hand, the CCE will induce a return effect. The areas with a development advantage will continue to develop and advance with their own favorable factors, which will aggravate the unbalanced development effect between regions and make the spatial development trend of polarization between high-value and low-value regions more and more obvious. On the other hand, the CCE will also cause diffusion effects. Capital, labor and other factors in developed areas will flow to the backward areas, thus promoting the overall high-quality development level of the region. 

The pattern of regional high-quality development has been different in different periods.

Specifically, in 2010, the overall level of high-quality development in the eastern coastal provinces was not high, and there were no higher areas and median areas. In terms of economic foundation, industrial structure, residents’ living standards and scientific and technological innovation, Shanghai had obvious initial advantages. Additionally, the spatial hierarchical distribution presented central symmetrical distribution characteristics, with Shanghai as the development leader. 

In 2012, the high-quality development level was significantly improved in the eastern coastal provinces. Especially with Shanghai as the center, the high-quality development level improved in stages in Shandong Province, Jiangsu Province, Zhejiang Province and Guangdong Province. This shows that Shanghai City on the one hand gave full play to its own characteristics and advantages to improve urban spatial structure and upgrade regional industrial structure. On the other hand, the city exported economic activities and production factors to surrounding areas, which promoted and stimulated the economic development of Jiangsu Province, Zhejiang Province and other districts.

In 2014, the pattern of spatial differences in high-quality development levels among provinces remained unchanged as a whole, but Hainan Province changed from a lower area to a low area. The levels of “economic vitality” and “innovation efficiency” have improved continuously in Hainan Province, but the levels of “residents’ living standards” and “green development” have decreased significantly. In particular, the water supply penetration rate for urban residents and the treatment rate for pollutants were lower significantly lower than the average.

In 2016, there was a new breakthrough in the high-quality development level of the eastern coastal provinces. With the respective development advantages of Jiangsu Province and Zhejiang Province and the leading role of Shanghai as a high-quality development leader, the economy, society, ecological environment and technological innovation developed in an comprehensive way, thereby forming a high-level area comprising Jiangsu, Shanghai, and Zhejiang.

In 2018, the level of high-quality development in the eastern coastal provinces significantly improved, mainly in peripheral areas such as Liaoning Province, Hebei Province, Fujian Province, and Guangdong Province, where the level of high-quality development has increased in stages. Relative to 2010, the high-quality development of the provinces and cities has moved in four stages, and the development gap between regions expanded due to the influence of backflow. There is still much room for high-quality development in the eastern coastal provinces.

In 2020, the high-quality development level continued to improve in the eastern coastal provinces. Shanghai City and Zhejiang Province were in the higher areas, and the advantages of high-quality development were still significant. Guangdong Province and Tianjin City were in high areas and becoming growth centers in the southern and northern zones of the eastern coastal provinces, respectively. Overall, high-quality development has moved forward and to a higher level in China’s eastern coastal provinces.

The development levels of the four dimensions of economic vitality, residents’ living standards, innovation efficiency and green development gradually improved in each province and city (Figure 8 and Table 4). At the same time, each province and city has different strengths and weaknesses, with obvious regional differences.

From these four dimensions of rankings for the provinces and cities, the green development indicators were more consistent with the ranking of high-quality development level, reflecting that green development is a key factor in achieving high-quality development. Provinces and cities with high levels of quality development, such as Shanghai, Jiangsu and Zhejiang, also had significantly better green development assessment values than other provinces and cities. Similarly, provinces with lower levels of quality development, such as Guangxi, Hebei and Liaoning, also had less green development assessment values than other provinces and cities. The scores for economic vitality, residents’ life, and innovation efficiency indicators were not the same as the level of high-quality development. Among them, especially the evaluation scores for residents’ life level had a large gap with the high-quality development level. For example, in provinces with a high-quality development level, such as Zhejiang Province and Tianjin City, the evaluation value of residents’ life indicators was at a low level. Contrarily, provinces a with low-quality development level, such as Guangxi Province and Hebei Province, had higher scores for residents’ life indicators.

From the four dimensions of high-quality development among areas, on the one hand, there was an obvious path dependence in the development level of each dimension in time; that is to say, the development mode continued to strengthen along the established direction in the provinces, and the development advantages did not change significantly. For example, the evaluation values of residents’ living standards have always had significant advantages in Liaoning Province, Hebei Province, Shandong Province, Guangxi Province, and Fujian Province, as they pay attention to people-oriented development concepts in the development process. Nevertheless, there were obvious deficiencies in the development of other dimensions. On the other hand, the development level of each dimension had a trickle-down effect in space, which fostered obvious improvements of different dimensions in neighboring provinces. For example, the evaluation value of the innovation efficiency dimension increased significantly in Shanghai City, Jiangsu Province and Zhejiang Province, and the evaluation value of residents’ living standards dimension improved significantly in Guangxi Province, Guangdong Province and Fujian Province. Therefore, the improvement in development quality will be affected by history in the provinces, and it is also inseparable from the interaction between regions.

## 4. Factor Analysis of Obstacles to High-Quality Development

### 4.1. Analysis of Overall Obstacle Factors in Eastern Coastal Provinces

#### 4.1.1. Obstacle Analysis of Four Dimensions of High-Quality Development Level

The obstacle degree of four dimensions of high-quality development level were calculated for the period 2010 to 2020 (Table 5). The barriers and transformation trends in the four dimensions were different. Economic vitality and residents’ living standards have been a great obstacle to high-quality development in the eastern coastal provinces, and the impact of economic vitality on high-quality development gradually increased during the study period. The obstacle of innovation efficiency decreased the most.

From the perspective of action intensity, the ranking of obstacle degree to high-quality development level in the four dimensions was basically stable. Specifically, the obstacle degree of residents’ living standards was the largest, followed by economic vitality and innovation efficiency, and the obstacle degree of green development was the smallest. This shows that residents’ living standards limited the high-quality development level in the eastern coastal provinces. As the focus of China’s economic development, the eastern coastal provinces are more likely to face development problems such as dense population, high employment pressure, large gaps between urban and rural areas and heavy burdens on infrastructure, which will conversely reduce people’s satisfaction with life.

From the change trend, the continuous improvement in residents’ living standards has reduced the obstacle degree of high-quality development, but economic vitality increasingly hinders high-quality development in the eastern coastal provinces. The largest reduction in barriers to innovation efficiency reflected that the level of scientific and technological innovation continued to reduce the obstacle to high-quality development, and barriers to green development were relatively stable.

#### 4.1.2. Analysis of Main Obstacle Factors of High-Quality Development Level

The obstacle degrees of 21 indicators of high-quality development in the eastern coastal provinces were calculated during the period 2010 to 2020. Due to the large number of indicators, this paper listed the top ten obstacle factors of high-quality development level from 2010 to 2020 (Table 6). If the obstacle degree of an obstacle factor was greater than 10%, that factor was a significant obstacle factor. It was found that the significant obstacle factors to high-quality development gradually changed from insufficient output in science and innovation to the reduction in the scale of foreign trade, and the problems of unreasonable industrial structure and shortages of per capita cultural resources persisted.

From the perspective of action intensity, X_14_ (proportion of total export–import volume in GDP), X_26_ (public library stock per capita), X_32_ (number of patent applications accepted per 10,000 people) and X_12_ (industrial structure advancement) have always been high obstacles to high-quality development in the eastern coastal provinces. Therefore, there were some development problems in the eastern coastal provinces, such as unreasonable industrial structure, insufficient output of innovation achievements and high population pressure, while weak consumer demand and insufficient investment in medical, education and social security facilities were also development problems that could not be ignored in the eastern coastal regions.

From the change trend, among the top ten obstacle factors of high-quality development levels in the eastern coastal provinces, the barrier degrees of all barrier factors were decreasing, except for X_26_ (public library stock per capita), X_14_ (proportion of total export–import volume in GDP) and X_31_ (FTE of R&D personnel per 10,000 people). In particular, the obstacle degree of X_14_ (proportion of total export–import volume in GDP) increased by 5.2%, the largest increase, while the obstacle degree of X_32_ (number of patent applications accepted per 10,000 people) decreased by 3.8%, for the largest decrease. Therefore, the shortage of public cultural resources, the shrinking scale of foreign trade, and the insufficient investment in innovation manpower increasingly affected the improvement of development quality in the eastern coastal provinces.

### 4.2. Obstacle Factor Analysis of Provinces in the Eastern Coastal Provinces

#### 4.2.1. Analysis of Obstacles in Four Dimensions of High-Quality Development Level in Provinces and Cities

We calculated the obstacle degree of four dimensions in the provinces in 2020 (Figure 9). The results show that provinces and cities with relatively developed economies also had low barriers in the four dimensions. However, the obstacle factors with the greatest degree of obstacles in different provinces and cities were different.

There were significant differences in the obstacle factors that comprised the largest obstacle degrees in different provinces. On the one hand, provinces with high-quality development levels and relatively developed economies had lower obstacle degrees than other, backward provinces, and the obstacle factors with the largest obstacle degrees were economic vitality and residents’ living standards. For example, the biggest obstacle factor was economic vitality in Jiangsu Province, Zhejiang Province, and Fujian Province, and the economic aggregate, industrial core competitiveness and opening-up level all affected regional economic vitality. The biggest obstacle factor was residents’ living standards in Shanghai City, Tianjin City and Guangdong Province, while the dense population, heavy pressure on infrastructure, medical and educational resources, and the prominent contradictions between man and land were the three key factors restricting high-quality development. On the other hand, the provinces with low levels of high-quality development and relatively backward economies had higher obstacle degrees than other, developed provinces, and the obstacle factors that were the largest obstacle degrees were green development and innovation efficiency. The biggest obstacle factor was green development in Hebei Province, Guangxi Province and Liaoning Province, while the low efficiency of pollutant treatment, the concentration of pollution-intensive enterprises, and the low utilization rate of resources were the three key factors that restricted high-quality development. Meanwhile, the biggest obstacle factor was innovation efficiency in Shandong Province and Hainan Province, while the lack of scientific and technological innovation capacity restricted the continuous improvement of economic development quality.

#### 4.2.2. Analysis of Main Obstacle Factors of High-Quality Development Level in Various Provinces

According to the obstacle degree model, the obstacle degree of 21 indicators in eastern coastal provinces in 2020 could be calculated, and this paper listed the top five obstacle factors (Table 7). The results show that the main obstacle factors had both similarities and differences, and the obstacle degrees were significantly different in the eastern coastal provinces.

On the whole, there were six common obstacle factors restricting high-quality development in the eastern coastal provinces. They were mainly X_14_ (proportion of total export–import volume in GDP; 11 times, with an average obstacle degree of 13.4%), X_26_ (public library stock per capita; 10 times, with an average obstacle degree of 12.5%), X_12_ (advanced industrial structure; 8 times, with an average obstacle degree of 12.2%), X_31_ (FTE of R&D personnel per 10,000 people; 7 times, with an average obstacle degree of 11.2%), X_28_ (ratio of urban to rural per capita disposable income; 7 times, with an average obstacle degree of 12.5%), and X_32_ (number of patent applications accepted per 10,000 people; 6 times, with an average obstacle degree of 12.0%), which appeared frequently and had a great obstacle degree. All provinces and cities were affected by these six obstacle factors, which belonged to common obstacle factors. This reflects the universal development problems in the eastern coastal provinces, including the reduction in the scale of foreign trade, the shortage of public cultural resources per capita, the unreasonable industrial structure, the large development gap between urban and rural districts, the insufficient investment in innovation, and the low output of scientific and technological innovation achievements.

From the perspectives of the provinces and cities, there were significant differences in the main obstacle factors and rankings. Except for X_25_ (number of full-time teachers per 100 general high school students), the top five obstacle factors were common obstacle factors in Tianjin City, Jiangsu Province and Zhejiang Province. In addition to the common obstacle factors, there were also two obstacle factors with low obstacle degrees in Shanghai, namely X_25_ (number of full-time teachers per 100 general high school students) and X_23_ (per capita road area). In addition to the six common barrier factors, Guangdong Province also has an obstacle factor with a large obstacle degree, X_24_ (number of health institutions per 10,000 people), which was quite different from other provinces and cities. The top five obstacle factors belonged to the common obstacle factors in Liaoning Province, Hainan Province, Shandong Province, Fujian Province, Guangxi Province and Hebei Province.

## 5. Conclusions

### 5.1. Major Findings

Taking the eastern coastal provinces as the research area, based on the goal of high-quality development, this paper evaluated the high-quality development level and analyzed the obstacle factors affecting high-quality development in the eastern coastal provinces by constructing an evaluation system and using the entropy-TOPSIS method. The main conclusions are as follows:(1)While GDP growth slowed down, the quality of development gradually improved in the eastern coastal provinces, as shown by the gradual increase from 3.212 in 2010 to 3.977 in 2012, then decreasing to 3.864 in 2014, and then gradually increasing to 4.979 in 2020, showing a “rise-decline-rise” upward fluctuation. In terms of the development of the four dimensions, the level of development of all four dimensions had been increasing, and the development level of the residents’ living standards dimension was the highest, at 2.440 in 2020, which was much higher than the development level of the other dimensions, while the level of development of the dimension of innovation efficiency had the fastest growth rate, at 66.1%. The level of high-quality development of the provinces showed different degrees of growth, while the coefficient of variation decreased from 0.331 to 0.271, gradually tending toward balanced development among the provinces. In Shanghai, Jiangsu Province, and Zhejiang Province, the high-quality development level was high and in a dominant position, while in Liaoning Province, the development level growth rate was 88.4%, the largest growth rate.(2)The spatial distribution pattern of the eastern coastal provinces in China has been relatively stable, showing a pattern of “high in the east and low in the west, high in the north and low in the south”, and the spatial differences in the north–south and east–west directions have been obviously increasing. From the spatial pattern of the four dimensions, the spatial evolution of each dimension was relatively stable, and generally showed a spatial distribution pattern with high-value areas as the core, among which the bipolar spatial effect of the innovation efficiency dimension was becoming more prominent, while the regional synergistic development effect of the resident living standards dimension was more obvious. From the spatial pattern of each province, there were significant differences among the provinces, showing the spatial distribution characteristics of “center–periphery” in general, in which the high-value areas were mainly Shanghai and Jiangsu, and the surrounding provinces belonged to the peripheral areas of high-quality development, and there was a CCE in the evolution of high-quality development in various provinces, which resulted in the high-quality development in various regions being affected by the return or diffusion effect for surrounding regions.(3)The greatest barriers to residents’ living standards and economic vitality were found in China’s eastern coastal provinces as a whole. The barriers to residents’ living standards decreased from 40.62% to 39.61%, but the barriers to economic vitality increased from 30.48% to 33.49%. In terms of the development of each barrier factor, the barrier factors with a barrier degree greater than 10% changed from three barrier factors (X_32_ number of patent applications per 10,000 people, X_12_ industrial structure advancement, X_26_ public library stock per capita) to two barrier factors (X_14_ proportion of total export–import volume in GDP, X_26_ public library stock per capita), which indicated that the significant obstacle factors gradually changed from insufficient output in innovation and unreasonable industrial structure to reduction in the scale of foreign trade and the problems and shortages of per capita public cultural resources.(4)Barrier factors in the eastern coastal provinces of China varied with the level of economic and quality development. For provinces with a high level of high-quality development and a relatively developed economy, the biggest obstacle factors were economic vitality and residents’ living standards, such as in Jiangsu, Zhejiang, Shanghai, and Guangdong provinces. For provinces with low levels of high-quality development and relatively backward economies, the biggest obstacle factors of obstacle degree were green development and innovation efficiency, such as in Hebei, Guangxi, Liaoning, and Hainan provinces. From the development of each obstacle factor, there were six common obstacle factors for high-quality development in the provinces of the eastern coastal region, which reflected common development problems in the region, mainly including the reduction in the scale of foreign trade, shortages of public cultural resources per capita, unreasonable industrial structure, large development gaps between urban and rural communities, insufficient investment in innovation and low output of scientific and technological innovation achievements. However, different provinces in the stage of high-quality development had different obstacle factors, such as Guangdong, which also had an obstacle factor with a large obstacle degree, X_24_ (number of health institutions per 10,000 people).

### 5.2. Planning Implications

Based on the results of this study on the high-quality development of the eastern coastal provinces, some rational suggestions for the high-quality development of the region are proposed:

Under a new development concept, high-quality development pays more attention to the improvement of economic quality and efficiency, both to achieve rapid economic development, while environmental protection, innovation-driven openness, coordination and sharing are all issues to be considered in the process of economic growth. The provinces should actively promote the overall development of all areas of economic vitality, innovation and efficiency, green development, and people’s living standards.

The improvement of development quality will be affected by the return effect or diffusion effect of the surrounding areas, so each province should clarify their own positioning in regional development, strengthen exchange and cooperation, and jointly solve common problems of development, such as unreasonable industrial structure, insufficient consumer demand and the insufficient infrastructure carrying capacity faced in regional development.

The improvement of development quality also will be affected by the culture, policies and other aspects in the original regional economic development model. Therefore, in order to achieve high-quality development, according to the development status and main obstacle factors in different regions, authorities need to adjust measures to local development problems. 

### 5.3. Limitations

A region is an open and complex giant system, and it is not enough to analyze the connotation of high-quality development only from the provincial scale in this paper. In the future, it will be necessary to analyze the spatial and temporal evolution of regional high-quality development levels from a more microscopic scale by combining multiple sources of data. Moreover, the spatial and temporal evolution of high-quality development is influenced by a variety of factors. In addition, in view of the current deficiencies in the high-quality development of the eastern coastal provinces, further research and studies on the high-quality development of the region can be conducted in the future to assess the formulation and implementation of policy goals in the region, and to provide a long-term tracking survey and policy guidance for its continuous and optimized development.

## Figures and Tables

**Figure 1 ijerph-20-01336-f001:**
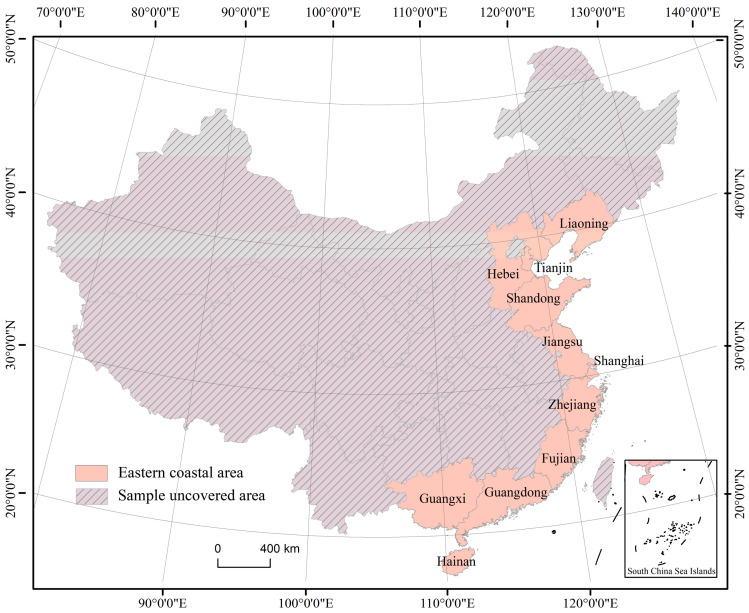
Regional overview of eastern coastal provinces.

**Figure 2 ijerph-20-01336-f002:**
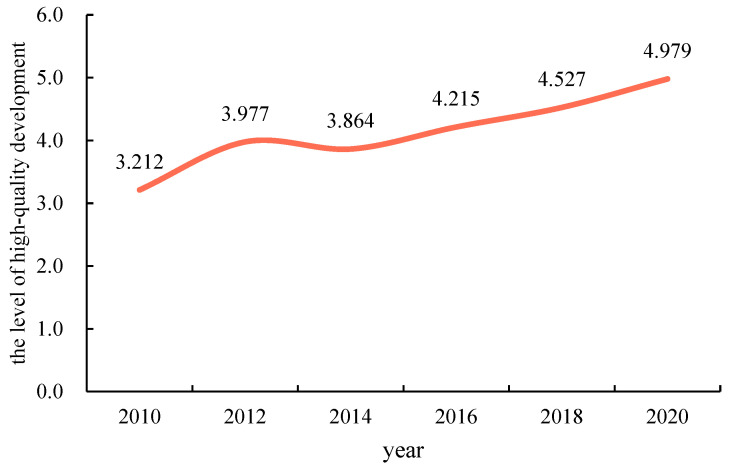
Changes in high-quality development in the study area from 2010–2020.

**Figure 3 ijerph-20-01336-f003:**
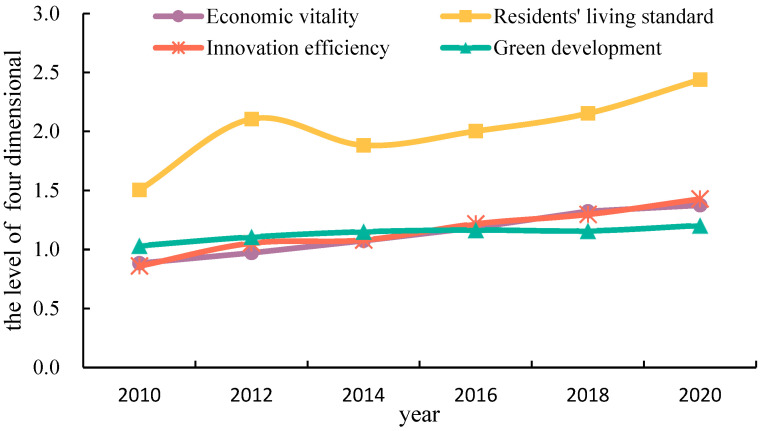
Changes in the four dimensions of high-quality development from 2010 to 2020.

**Figure 4 ijerph-20-01336-f004:**
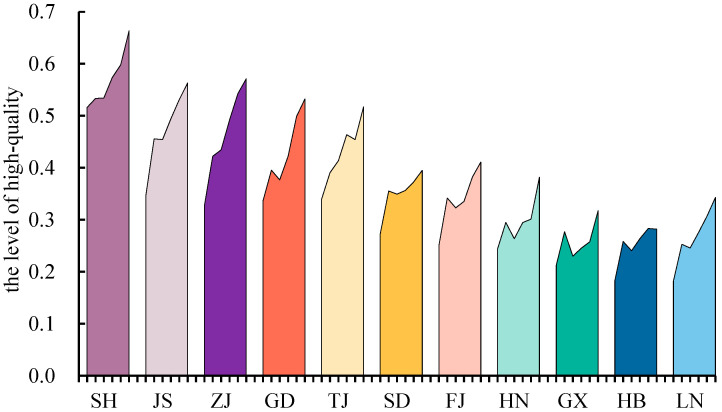
Development and changes of high quality in provinces during 2010–2020. SH, JS, ZJ, GD, TJ, SD, FJ, HN, GX, HB, LN mean Shanghai City, Jiangsu Province, Zhejiang Province, Guangdong Province, Tianjin City, Shandong Province, Fujian Province, Hainan Province, Hebei Province, Guangxi Province and Liaoning Province.

**Figure 5 ijerph-20-01336-f005:**
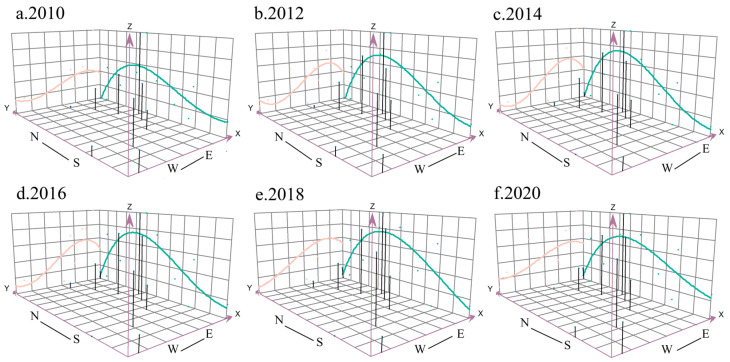
Spatial trends in the level of high-quality development from 2010 to 2020. N, S, W and E mean north, south, west and east, respectively.

**Figure 6 ijerph-20-01336-f006:**
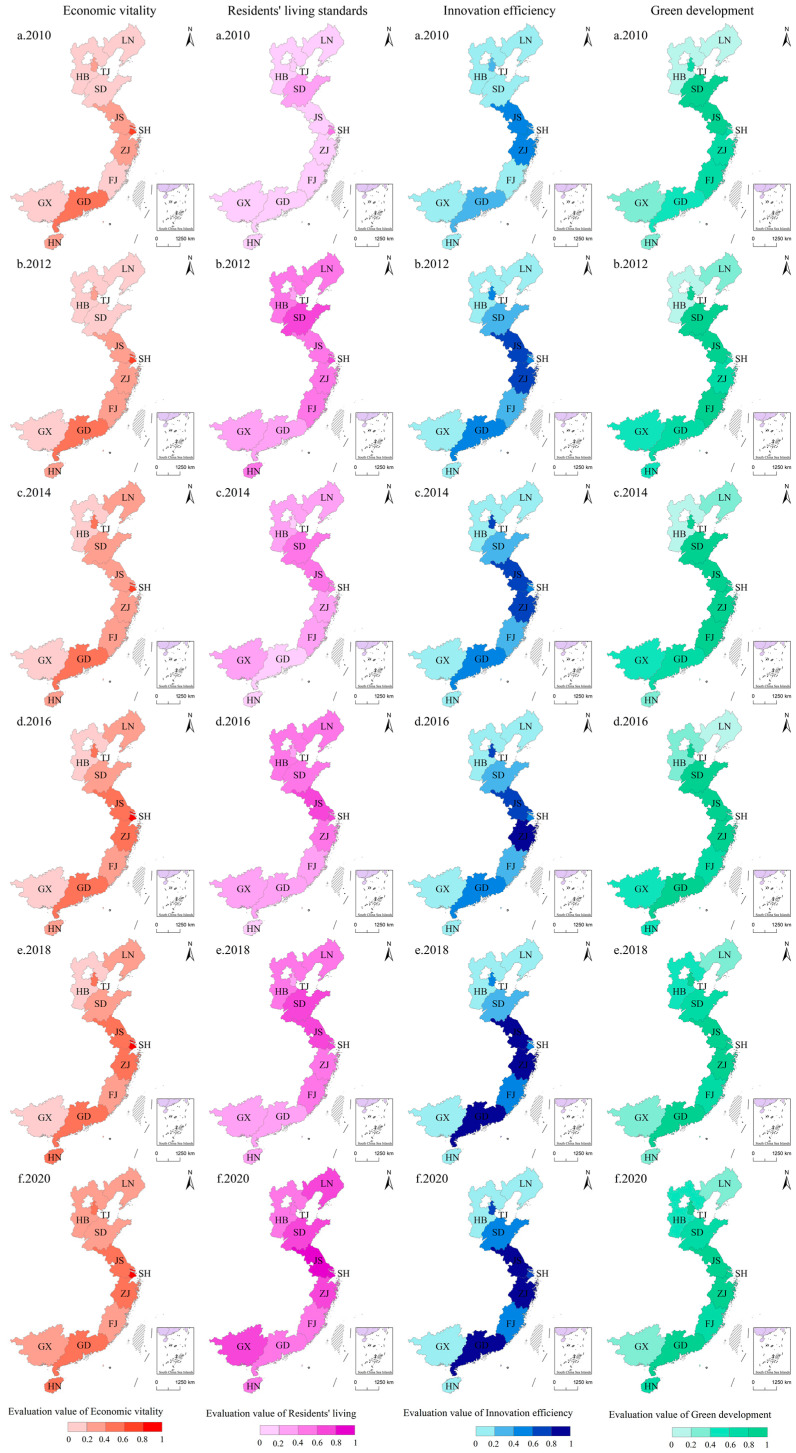
Spatial pattern of development levels in each dimension from 2010 to 2020.

**Figure 7 ijerph-20-01336-f007:**
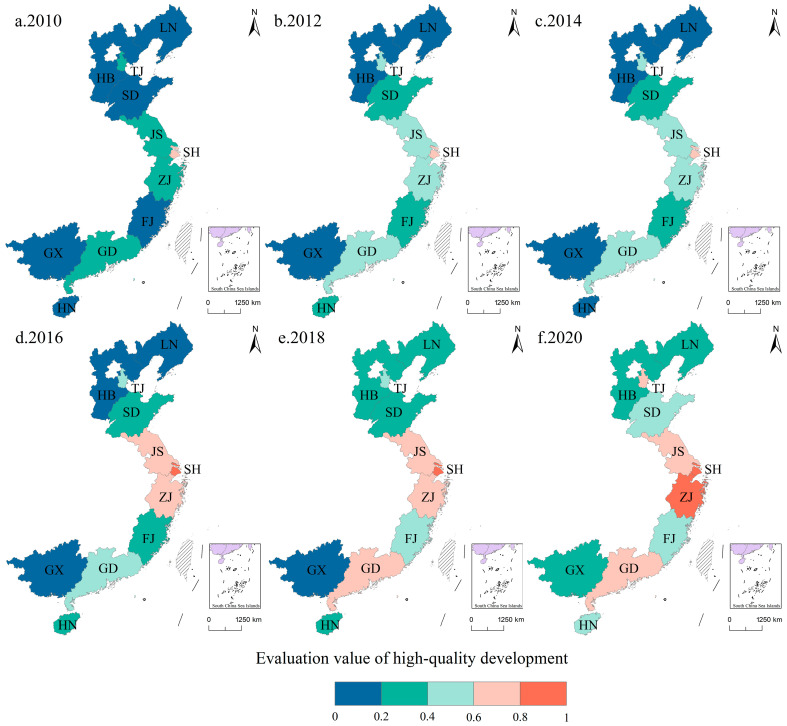
The spatial distribution pattern of high-quality development levels, 2010–2020.

**Figure 8 ijerph-20-01336-f008:**
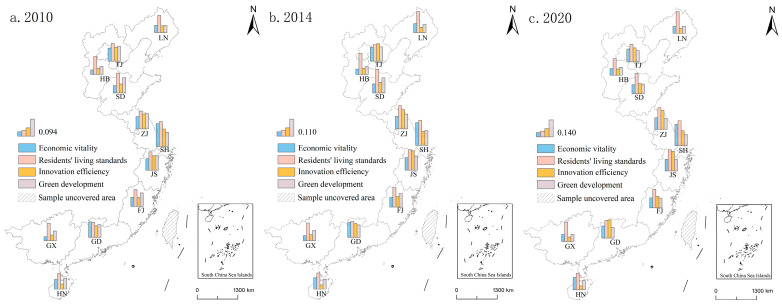
Spatial evolution of the four dimensions of high-quality development level, by province and city.

**Figure 9 ijerph-20-01336-f009:**
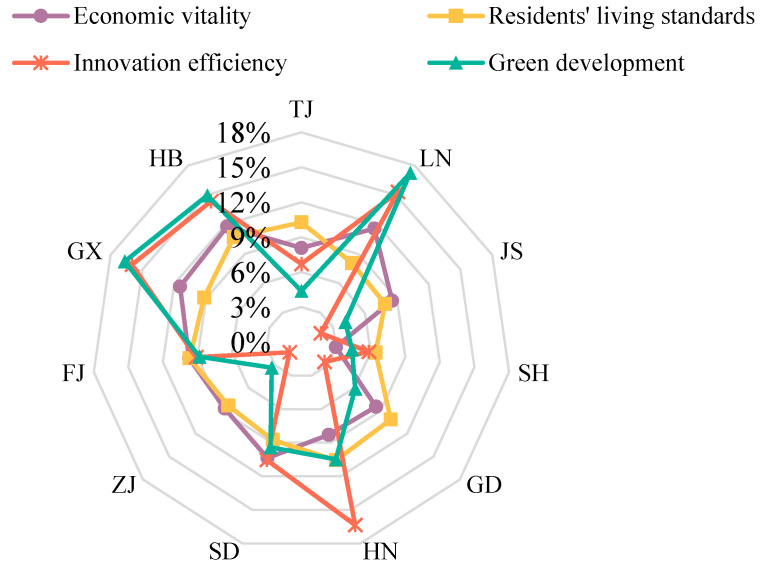
The obstacle degrees of high-quality development criteria in eastern coastal provinces in 2020.

**Table 1 ijerph-20-01336-t001:** Index system for evaluation of high-quality development level.

Dimensions	Indicators	Company	Index Attribute
Economic vitality X_1_ (0.270)	Per-capita GDP, X_11_ (0.050)	CNY	+
	Industrial structure advancement, X_12_ (0.088)	p.c	+
	Total retail sales of consumer goods per capita, X_13_ (0.082)	CNY	+
	Proportion of total export–import volume in GDP, X_14_ (0.050)	P.c	+
Residents’ living standards X_2_ (0.387)	Engel’s coefficient of residents, X_21_ (0.026)	P.c	-
	Coverage rate of endowment insurance, X_22_ (0.038)	P.c	+
	Per capita road area, X_23_ (0.059)	m^2^	+
	Number of health institutions per 10,000 people, X_24_ (0.088)	-	+
	Number of full-time teachers per 100 general high school students, X_25_ (0.008)	-	+
	Public library stock per capita, X_26_ (0.051)	-	+
	Urban water penetration rate, X_27_ (0.046)	P.c	+
	Ratio of urban to rural per capita disposable income, X_28_ (0.071)	-	-
Innovation efficiency X_3_ (0.213)	FTE of R&D personnel per 10,000 people, X_31_ (0.035)	year	+
	Number of patent applications accepted per 10,000 people, X_32_ (0.078)	-	+
	Proportion of science and education expenditure in general budget expenditure, X_33_ (0.100)	P.c	+
Green development X_4_ (0.130)	Treatment rate for urban sewage, X_41_ (0.018)	P.c	+
	SO_2_ emissions per CNY 10,000 of industrial output value, X_42_ (0.009)	Tons of standard coal/CNY 10,000	-
	Power consumption per CNY 10,000 of GDP, X_43_ (0.010)	Tons/CNY	-
	Greening coverage rate of built-up area, X_44_ (0.020)	P.c	+
	Utilization rate of general industrial solid waste, X_45_ (0.051)	P.c	+
	Treatment rate for MSW, X_46_ (0.022)	P.c	+

**Table 2 ijerph-20-01336-t002:** Regional differences in quality development from 2010 to 2020.

	2010	2012	2014	2016	2018	2020
CV	0.331	0.245	0.288	0.291	0.289	0.271
Range	0.334	0.281	0.304	0.328	0.340	0.382

**Table 3 ijerph-20-01336-t003:** Regional quantitative changes in high-quality development stage during 2010–2020.

Year	2010	2012	2014	2016	2018	2020
higher area	0	0	0	1	1	2
high area	1	1	1	2	3	3
median area	0	4	4	2	2	3
low area	4	3	2	3	4	3
lower area	6	3	4	3	1	0

**Table 4 ijerph-20-01336-t004:** The four dimensions of high-quality development level, by province and city.

Year	Dimension	TJ	LN	JS	SH	GD	HN	SD	ZJ	FJ	GX	HB	SUM	CV
2010	Economic vitality	0.098	0.055	0.093	0.173	0.118	0.073	0.055	0.087	0.069	0.030	0.033	0.884	0.508
	Residents’ living standards	0.134	0.130	0.134	0.189	0.116	0.118	0.150	0.138	0.127	0.133	0.136	1.504	0.143
	Innovation efficiency	0.104	0.051	0.113	0.133	0.087	0.039	0.068	0.109	0.067	0.044	0.046	0.860	0.418
	Green development	0.113	0.054	0.117	0.106	0.097	0.083	0.116	0.110	0.103	0.074	0.060	1.032	0.242
2012	Economic vitality	0.109	0.064	0.099	0.182	0.123	0.080	0.066	0.095	0.077	0.036	0.038	0.971	0.468
	Residents’ living standards	0.165	0.186	0.204	0.225	0.161	0.186	0.221	0.193	0.192	0.179	0.195	2.106	0.105
	Innovation efficiency	0.122	0.056	0.164	0.121	0.118	0.039	0.089	0.152	0.087	0.049	0.056	1.054	0.446
	Green development	0.122	0.069	0.117	0.112	0.101	0.091	0.122	0.113	0.115	0.089	0.055	1.106	0.222
2014	Economic vitality	0.118	0.075	0.107	0.194	0.128	0.097	0.077	0.107	0.083	0.044	0.045	1.074	0.429
	Residents’ living standards	0.145	0.179	0.198	0.213	0.138	0.138	0.201	0.175	0.166	0.152	0.179	1.883	0.152
	Innovation efficiency	0.149	0.045	0.163	0.118	0.120	0.036	0.089	0.163	0.091	0.053	0.052	1.080	0.487
	Green development	0.121	0.069	0.126	0.125	0.107	0.083	0.128	0.120	0.118	0.088	0.065	1.151	0.227
2016	Economic vitality	0.128	0.092	0.119	0.206	0.134	0.108	0.087	0.118	0.087	0.050	0.055	1.185	0.396
	Residents’ living standards	0.157	0.193	0.215	0.226	0.150	0.145	0.208	0.188	0.171	0.163	0.185	2.003	0.150
	Innovation efficiency	0.164	0.049	0.182	0.132	0.142	0.035	0.098	0.191	0.107	0.057	0.058	1.217	0.505
	Green development	0.123	0.064	0.125	0.126	0.119	0.092	0.115	0.120	0.107	0.089	0.084	1.165	0.194
2018	Economic vitality	0.141	0.100	0.134	0.221	0.142	0.119	0.099	0.136	0.101	0.065	0.065	1.323	0.361
	Residents’ living standards	0.160	0.210	0.229	0.237	0.163	0.157	0.224	0.204	0.193	0.179	0.199	2.154	0.144
	Innovation efficiency	0.144	0.050	0.194	0.139	0.193	0.035	0.097	0.215	0.126	0.048	0.057	1.297	0.554
	Green development	0.125	0.074	0.129	0.123	0.122	0.078	0.111	0.123	0.105	0.081	0.087	1.158	0.201
2020	Economic vitality	0.144	0.078	0.135	0.241	0.137	0.140	0.101	0.132	0.113	0.080	0.074	1.375	0.374
	Residents’ living standards	0.195	0.245	0.249	0.281	0.198	0.189	0.230	0.239	0.208	0.220	0.186	2.440	0.135
	Innovation efficiency	0.158	0.058	0.214	0.168	0.205	0.045	0.113	0.220	0.129	0.050	0.069	1.428	0.522
	Green development	0.126	0.080	0.127	0.126	0.120	0.104	0.108	0.130	0.110	0.082	0.088	1.204	0.171

**Table 5 ijerph-20-01336-t005:** Obstacles to high-quality development level of subsystems in eastern coastal provinces from 2010 to 2020.

Year	Economic Vitality	Residents’ Living Standards	Innovation Efficiency	Green Development
2010	30.48%	40.62%	22.12%	6.79%
2012	33.47%	37.50%	22.37%	6.66%
2014	31.74%	40.46%	21.84%	5.96%
2016	31.90%	41.00%	21.02%	6.09%
2018	31.53%	40.95%	20.95%	6.57%
2020	33.49%	39.61%	20.54%	6.35%

**Table 6 ijerph-20-01336-t006:** Ranking of the top ten obstacle factors of high-quality development level in China’s coastal provinces from 2010 to 2020 (%).

Year	Ranking	1	2	3	4	5	6	7	8	9	10
2010	Obstacle factor	X_32_	X_12_	X_26_	X_22_	X_14_	X_31_	X_25_	X_11_	X_13_	X_28_
	Obstacle degree (%)	12.08	10.72	10.29	8.06	7.87	7.66	6.72	6.05	5.84	5.50
2012	Obstacle factor	X_32_	X_12_	X_26_	X_14_	X_31_	X_25_	X_11_	X_13_	X_22_	X_24_
	Obstacle degree (%)	12.21	11.96	10.83	9.32	8.58	7.38	6.17	6.02	5.76	5.41
2014	Obstacle factor	X_32_	X_12_	X_26_	X_14_	X_31_	X_28_	X_25_	X_22_	X_11_	X_13_
	Obstacle degree (%)	11.73	11.13	10.96	9.76	7.78	7.54	6.82	5.72	5.56	5.29
2016	Obstacle factor	X_26_	X_14_	X_12_	X_32_	X_31_	X_28_	X_25_	X_22_	X_11_	X_13_
	Obstacle degree (%)	11.11	11.10	10.50	9.98	8.21	8.10	6.85	6.05	5.38	4.91
2018	Obstacle factor	X_14_	X_26_	X_12_	X_32_	X_28_	X_31_	X_25_	X_22_	X_11_	X_13_
	Obstacle degree (%)	11.36	11.21	10.52	9.24	8.68	8.55	7.01	6.35	4.97	4.68
2020	Obstacle factor	X_14_	X_26_	X_28_	X_12_	X_31_	X_32_	X_11_	X_13_	X_25_	X_22_
	Obstacle degree (%)	13.06	11.67	9.90	9.66	8.94	8.29	5.45	5.31	5.18	4.95

**Table 7 ijerph-20-01336-t007:** Ranking of the top five barriers to high-quality development in China’s coastal provinces in 2020 (%).

Region	Ranking	1	2	3	4	5
TJ	Obstacle factor	X_28_	X_14_	X_25_	X_26_	X_31_
	Obstacle degree (%)	12.43	11.79	11.79	10.27	9.50
LN	Obstacle factor	X_32_	X_14_	X_31_	X_26_	X_12_
	Obstacle degree (%)	12.07	11.43	10.22	9.97	9.16
JS	Obstacle factor	X_12_	X_14_	X_26_	X_28_	X_25_
	Obstacle degree (%)	17.56	17.43	15.64	13.82	7.30
SH	Obstacle factor	X_28_	X_14_	X_31_	X_25_	X_23_
	Obstacle degree (%)	18.25	14.93	14.88	13.67	13.08
GD	Obstacle factor	X_26_	X_12_	X_14_	X_28_	X_24_
	Obstacle degree (%)	15.37	13.25	11.00	10.26	6.72
HN	Obstacle factor	X_32_	X_14_	X_31_	X_26_	X_28_
	Obstacle degree (%)	13.61	12.87	12.64	12.13	8.03
SD	Obstacle factor	X_26_	X_14_	X_32_	X_12_	X_31_
	Obstacle degree (%)	13.59	13.25	11.28	11.07	9.75
ZJ	Obstacle factor	X_14_	X_12_	X_28_	X_26_	X_25_
	Obstacle degree (%)	16.67	16.06	15.01	13.24	7.17
FJ	Obstacle factor	X_14_	X_12_	X_26_	X_32_	X_28_
	Obstacle degree (%)	14.15	13.40	11.56	9.50	9.34
GX	Obstacle factor	X_32_	X_14_	X_26_	X_31_	X_12_
	Obstacle degree (%)	13.35	11.96	11.54	11.42	7.83
HB	Obstacle factor	X_32_	X_26_	X_14_	X_31_	X_12_
	Obstacle degree (%)	11.91	11.88	11.84	9.97	8.85

## Data Availability

Not applicable.

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
