# Peer review of "Evaluation and Impact Mechanism of High-Quality Development in China’s Coastal Provinces"

_ijerph, 2023, doi:10.3390/ijerph20021336_

Round 1
Reviewer 1 Report
Taking China’s eastern coastal areas as an example, this study uses the entropy-TOPSIS method and obstacle degree model to explore spatio-temporal evolution characteristics of high-quality development and obstacle factors in coastal China. Generally the paper is well written. However, there are some concerns as stated below:
1. Some references are missed in the manuscript, for example,“literature on research techniques”,“literature on the eastern coastal regions that is relevant” and “literature on high-quality development evaluation”.
2. I would like to suggest separate the discussion from the results, and the discussion should include specific suggestions for promoting high-quality regional development in the region.
3. Some Figures can be further improved. For example, you should use the same color system in the legend of Figure 1.
4. I would recommend a thorough language editing by an Enlish-speaking researcher. Also, some statements are not consistent, e.g., ”high-quality development ”and “high quality development”.
Reviewer 2 Report
I think that there is a problem with the coastal area from a geographical point of view.
In Europe when we speak about the coastal area we intend an area constituted by municipally with almost 50% of the surface in a belt of 10 km from the coastline.
In this work, the area is constituted of counties that are very large.
So my first question is but are we sure that we can speak about the coastal area?
I think that the authors can explain better the reason for this choice.
Simply the reason may be that we have data only for this granulometry.
Obviously, if they have data at the highest resolution they must take a report between this area and a subset more close to the coastline.
I'd like to know if they can have these data to check the problem in a really coastal area.
Round 2
Reviewer 2 Report
Great good job.
Allow me just a small comment.
I find very appropriate the title of Conclusion paragraph :" Conclusion and planning implications"
I would suggest that you point out that the choice of coastal provinces is due to the fact that the province is an administrative body that can make decisions. This justifies the choice to investigate this area also in the future in order to judge the achievement of the objectives of the policy at local scale.
Great!
Best regard and Happy Xmas!
Author Response
请参阅附件
